# Teachers' Satisfaction, Role, and Digital Literacy during the COVID-19 Pandemic

**Ming Li and Zhonggen Yu \***

Faculty of Foreign Studies, Beijing Language and Culture University, Beijing 100083, China;
202121296091@stu.blcu.edu.cn
\* Correspondence: yuzhonggen@blcu.edu.cn

**Abstract:** The COVID-19 pandemic has unexpectedly affected the educational process worldwide, forcing teachers and students to transfer to an online teaching and learning format. Compared with the traditional face-to-face teaching methods, teachers' professional role, career satisfaction level, and digital literacy have been challenged in the COVID-19 health crisis. To conduct a systematic review, we use critical appraisal tools from the University of the West of England Framework We removed the irrelevant and lower-quality results to refine the results and scored each selected paper to get high-quality studies with STARLITE. The number of finally included studies is 21. We used the PICO mnemonic to structure the four components of a clinical question, i.e., the relevant patients or population groups, the intervention (exposure or diagnostic procedure) of interest, as well as against whom the intervention is being compared and considered appropriate (outcomes). We formulated five research questions regarding teachers' professional role, satisfaction, digital literacy, higher educational practice, and sustainable education. The study found that teachers' professional roles changed complicatedly. Moreover, they were assigned more tasks during the online teaching process, which also implicated a decline in teachers' satisfaction. After the COVID-19 pandemic, it is necessary to conduct a blended teaching model in educational institutes. Teachers should have adequate digital literacy to meet the new needs of the currently innovative educational model in the future. In addition, the study reveals that teachers' digital literacy level, career satisfaction, and professional role are significantly correlated. We measured to what degree the three factors affected the online teaching and learning process. Ultimately, the study may provide some suggestions for methodological and educational strategies.

**Keywords:** COVID-19; teacher professional role; teacher career satisfaction; teacher digital literacy

## 1. Introduction

The year 2020 has witnessed the urgent need for online higher education, which was courageous in redesigning education fundamentals [1]. Numerous schools made prompt arrangements for full online provision. The implications of online higher education have gained significant prominence. Online higher education aims to get educators into the Internet space, expanding students' learning opportunities [2]. When the online education market rapidly developed within a few decades, higher education institutions adapted to online course offering and design. Higher education with online technology may be a popular mode worldwide. Meanwhile, online technology drew numerous participants to use the online learning opportunities. These have prompted us to consider online higher education agents [1].

The COVID-19 pandemic has unexpectedly affected the educational process world-wide, forcing teachers and students to transfer to an online teaching and learning format. Educators and students had to gradually adapt to the digital educational platform, which is a tremendous challenge for all participants. Undoubtedly, online teaching during the COVID-19 pandemic or current half-virtual education format will bring about professional

role changes, career satisfaction alterations, and new requirements for teachers' digital literacy [3]. There have been extensive studies on these issues before the COVID-19 health crisis; research on these topics has never stopped. What new research directions are there in the pandemic with the virtual teaching format?

Compared with the traditional face-to-face teaching methods, teachers' professional role, career satisfaction level, and digital literacy have been challenged in the COVID-19 health crisis. Teachers' professional role and their satisfaction level play a crucial role that can affect the completion of curriculums. Meanwhile, remote teaching relied more on computer technology, which profoundly impacted reconstructing education during and after the COVID-19 pandemic [4]. The present study tends to explore research areas of teachers' professional role, satisfaction, and digital literacy.

Teachers must have adequate digital literacy to teach online, which is required in the current educational model. Nevertheless, digital literacy alone could not facilitate the teaching process [3]. A successful teaching process also involves teachers' professional roles and satisfaction. The three elements are intertwined and essential for overall online teaching and learning [4]. The present study explores the changes in teachers' professional role and career satisfaction levels and the challenges to their digital literacy during the COVID-19 pandemic worldwide. The combination of the three elements may enable teachers to perform duties in schools better. Moreover, the present study explores the changes of the role of online higher education as an active agent during the COVID-19 pandemic, because teachers and students are mainly dependent on online-technology-based platforms to sustain their education.

As frontline providers of education, teachers are increasingly important in educational settings, especially in the virtual teaching and learning environments. The COVID-19 pandemic accelerated the process of educational virtualization [5]. It seems that education at all levels developed online virtual learning platforms [5]. Teachers' profession can be regarded as a motivator to use virtualization in teaching, where many different aspects of the teaching process are connected. The teacher's professional role, as a pedagogue, can solve students' problems [6]. A teacher's career satisfaction is a pleasant mental state arising from their appreciation of their work or experience [6]. It is important for teachers to feel satisfied with their work or profession. Teachers' digital literacy indicates the ability to use digital resources and virtual learning platforms in the educational environment. Teachers equipped with basic digital literacy will be highly competitive in future online or classroom practice [1]. Overall, as a knowledge transmitter, the teacher plays a significant role in virtual educational settings during the COVID-19 pandemic.

## 2. Literature Review

### 2.1. Teachers' Professional Role

A teacher's professional role embodies a multitude of implications [7]. Generally, its meaning has improved through professional experience over time [8]. The concept of a teacher's professional role is dynamic and is formed and reformed through time. The teacher's professional role is indispensable to education. Without the professional role of teachers, school activities are difficult to be carried out effectively. Even the professional role of a teacher makes a fundamental contribution to institutions and students [9]. With the increasing development of digital technology, teachers' professional role increased owing to online teaching and blended learning.

The role of a teacher involves many subject areas. The present study mainly focused on language education and found that teacher's professional role has constantly enriched the learning community over time [10]. It is beneficial to improve learning and teaching effectively, leading to a thriving learning community [11]. As a mediator, the teacher's professional role contributes to the students' language development; as a key figure, the teacher innovates second language education; as a change agent, the teacher actively contributes to pedagogy [12]. In in-person teaching practice, the teacher focuses on students' learning [13]. Previous studies in professional teacher roles have not dealt with current

situations and mainly focused on student-centered teaching. A study concerned the effects of the COVID-19 pandemic on teachers and their role changes because most schools and universities have switched to online learning [14]. The teaching environment for teachers and students has changed. This study proposes the research question:

RQ1: What changes has teachers' professional role experienced during the COVID-19 pandemic worldwide?

## 2.2. Teacher's Career Satisfaction

A teacher's career satisfaction is the realization of the requirements of the working environment [11]. According to some prospects, career satisfaction refers to employees' psychological and physical satisfaction with the work environment and the work itself, and it also indicates a worker's subjective reflection of the work situation [15]. Therefore, to determine the employees' satisfaction with work, there are five dimensions: the job itself, management, salary, promotion, and work partners [12]. Nevertheless, dissatisfaction is a negative emotion [16], resulting in an unsatisfactory career. In recent years, there has been an increasing amount of literature on teacher career satisfaction. Some studies have begun to examine teachers' career satisfaction, an enjoyable and active emotional state that originates from a person's job experience. Teacher's satisfaction is critical to institutions and organizations [17,18].

Some studies have begun to examine teachers' career satisfaction recently [19]. Some authors have mainly been interested in teachers' career satisfaction concerning predictors, factors, and levels [20–26]. Others measured the influences of both the extrinsic (contextual and organizational) and intrinsic (interpersonal or personal) factors [27]. Moreover, some studies addressed teachers' identification (occupational and organizational) [28]. However, some authors highlight teachers' general career satisfaction with life, job satisfaction, and emotional intelligence [29].

Moreover, much literature indicates the different research directions of teachers' career satisfaction. Some literature emphasizes teachers' leadership [30]. Other research directions are mainly about the teacher-rated school climate [31], working conditions, school organizational culture, the teacher's learning environment [32], the prevalence of low back pain (LBP) [33], the quality of mathematics textbooks [34], social services and the facilities of faculties and departments [35,36], leadership styles of academic deans [37], language teachers' identities [38], and the racial composition [39]. Furthermore, some researchers have found triggers for teacher career satisfaction. For example, some studies point out three decisive roles in teacher career satisfaction, i.e., competence for work, decision-making autonomy, and interpersonal and psychological aspects [40].

However, there are few published studies describing the leading role of teacher careers' satisfaction. It makes an important impact on a school's culture, language teaching [41], education, and students' academic achievements [42]. It also plays an important role in helping children with mental illness, providing more efficient professional performances, rendering a positive attitude toward work and life [43], and implementing specific programs [44]. Furthermore, some studies offer a somewhat different view of teachers' career dissatisfaction [45,46].

Overall, these studies highlight the importance of teachers' career satisfaction for future research and methodological diversity [47]. Given all that has been mentioned so far, the epidemic has given birth to new teaching methods. Will the previously unsatisfactory aspects become satisfying? The present study proposed the research question:

RQ2: Are teachers satisfied with their teaching profession during the COVID-19 pandemic worldwide?

## 2.3. Development of Teachers' Digital Literacy

In earlier research, teachers' digital literacy development became a vital issue in teaching and research [48]. The research on the improvement of teachers' skills and affordance of the necessary conditions in the digital era has become a hot issue [49]. Digital

literacy is a potential competence to realize teachers' professional development in e-learning communities confronting the global economy and computer technologies [50]. Moreover, successful individual teacher professional development requires adequate digital literacy in the fulfillment of e-learning environments [51]. Teachers' digital literacy often refers to the necessary skills and knowledge of teachers to learn in the digital knowledge society [52]. Digital tools are required for teachers to use information and communication technologies, manage information, and perform tasks collaboratively, effectively, and efficiently [53]. Usually, teachers' digital literacy development will enrich traditional education and create an immense vitality in complex organizational systems contexts [54]. The better teachers' digital literacy development, the more competent teachers are in education [55].

Recent studies suggested that the effective improvement of teachers' digital literacy in schools profoundly changed the conventional teaching and learning society [56]. Nevertheless, along with the advent of digital technology, there seemed to be risks of digital inequalities [57]. Many factors could affect the development of teachers' digital literacy. Teachers could be reluctant to acquire digital skills in existing professional development models at all times, especially in teacher-centered conventional classrooms. Some teachers deemed digital literacy training laborious and time-consuming and felt challenged to receive a digital competence within existing professional development programs [58]. Thus, the willingness to accept digital educational training dropped. The lack of school administrators to participate in the available teaching support was also an obstacle to a total digital transformation [59]. Currently, researchers are mainly focusing on institutional and peer exchanges rather than the development of networks and pedagogical skills. Moreover, digital literacy levels could be hindered by the support and school electronic environment. To some degree, the development of teachers' digital codependence depended on the readiness of school academic staff [60]. Teachers' readiness more or less counted on the flexibility of the school platform.

However, teachers' digital learning has accelerated owing to online teaching during the COVID-19 pandemic. The effective use of digital technology in schools also requires profound changes in teachers' digital literacy [56]. The development of teachers' digital literacy will deepen within blended teaching in the future [58]. With the current situation of the COVID-19 pandemic, new digital literacy skills have become indispensable to most education that is carried out through online communities and networks. In the long run, digital tools will integrate into conventional teaching, which will result in a sustainable reshaping of teachers' digital literacy. The present study reflected on the literature and proposed the research question:

RQ3: What is the role of digital literacy during the COVID-19 pandemic worldwide?

### 2.4. Online Higher Education

Online higher education refers to a revolutionary solution to different and unequal educational issues [1]. The adoption of online higher education would become beneficial to university education. Furthermore, higher education should fully take the form of online delivery [1]. At the early stage of online higher education, it aimed to offer non-traditional students access to educational opportunities. Current studies acknowledged that online higher education became increasingly complicated and diverse in practice [61], especially when it encompassed traditional face-to-face teaching and learning formats and technology-based online education. Some studies revealed that online higher education, an active agent, was interactive and collaborative in that it offered significant advantages over the conventional face-to-face style [1].

However, most positive descriptions of online higher education might depend on technological promises rather than on real-life online higher educational practices based on didactics [62]. Focusing on the programmatic definitions of online higher education might fail to develop and practice the theory during the COVID-19 pandemic [62]. In addition, some studies have demonstrated that higher dropout rates have been increasing for some time during the COVID-19 pandemic. Students were frequently frustrated due to a lack of

higher meta-cognitive skills, insufficient feedback from their instructors, and difficulties with distance learning [63]. Moreover, some literature considered online higher education a passive agent, tarnishing higher education [64]. Thus, the present study proposed the research question:

RQ4: Can online higher education be an active agent for the change?

### 2.5. Online Technology and a Sustainable Education

Sustainable higher education refers to the ability of universities to meet the specific needs of industries. Universities might collaborate with the industry to fulfill its needs and expectations because very few universities could produce employable and job-ready graduates on their own [65]. Nevertheless, a university could not collaborate with industries because of economic, cultural, social, or time constraints. There was a need for a mechanism that was effective and better to fulfill the expectations of the industry [65]. Meanwhile, a complete face-to-face delivery mode had its disadvantages. Online technology would deliver online higher education. Studies revealed that it would reduce or address the challenges through online technology [65]. Some research revealed that online technology might effectively help deliver online teaching and learning, benefiting the society [66]. Online technology is also a helpful logistical resource for in-class study.

However, some studies were concerned with the negative impact of online technology for its being an insufficient substitute for classroom teaching [67]. Online technology should be delivered and used alongside other means in different situations [67]. The knowledge delivery and skills output needed practice and teaching [67]. The efficient delivery of knowledge and skills could be realized by means of different teaching tools. An online implementation might not be limited to online technology. It was risky to take a complete online education shift [64]. Sustainable education challenged students psychologically via online technology in a developing society. The problem might become a major issue because students were unlikely to ask questions due to cultural norms, forming a gap between them and teachers. The presence of online technology might widen the gap during the COVID-19 pandemic [64]. Furthermore, the lack of a strong participatory learning culture in emerging societies might threaten students' academic success and job readiness [64].

Undeniably, online technology-assisted sustainable education was generally associated with employment [68]. Sustainable education made graduates have employable skills and academic success, which gave them job readiness [68]. Students' academic scores were considered a predictor to identify the success of their education. Although sustainable education might cater to market popularity, online technology could ensure that the right knowledge and skills were put first for different modes of technology occurring in different situations [69]. Online technology could be useful for academic success, as it could easily make all students focus on online learning [2]. However, as the COVID-19 pandemic continued to spread, there was a concern that technology-based sustainable education could merely be accepted by students with a high socio-economic status. Furthermore, it could not guarantee a qualified and proper education. Thus, it could not be promised that graduates could be successful in their academic career and job readiness due to its time consumption [2]. Thus, the present study proposed the research question:

RQ5: Does online technology provide a sustainable education in terms of academic success and job readiness before and during COVID-19?

## 3. Methodology

### 3.1. Literature Search

The search aims to analyze the emerging topics or research areas systematically. The author searches literature using all fields to ensure that all papers are related to the same field. The first search strings for the present study were COVID-19* pandemic AND "teacher* AND role OR roles OR teacher* role OR "teacher role*" OR "teacher* role" OR "teacher role" OR professional role". The second search strings were COVID-19* pandemic AND teacher* AND career satisfaction OR "teacher career satisfaction" OR

"teacher* satisfaction OR "teacher satisfaction*". The third search strings were COVID-19* pandemic AND teacher* AND digital literacy AND development OR "teacher digital literacy development" OR "teacher* digital literacy development" OR "teacher digital literacy development*". Documents were searched for across the Web of Science database in November 2021 and restricted by publishing year (2020–2021), and the search brought up 662 articles.

### 3.2. Main Inclusion Criteria

We firstly screened the articles based mainly on the appropriateness of abstracts. Secondly, we included the articles if they fulfilled the following conditions: (1) the publications were made available in the form of articles; (2) the empirical study demonstrated teachers' satisfaction, role, and digital literacy during the COVID-19 pandemic; (3) the articles focused on educational domains; (4) the studies were written in the English language; (5) the dependent variables should be teachers' satisfaction, role, or digital literacy; and (6) the studies provided sufficient data and analyses from which the conclusions were drawn (e.g., standard deviations or means). Based on the inclusion criteria, 21 articles were included in the present study (Figure 1).

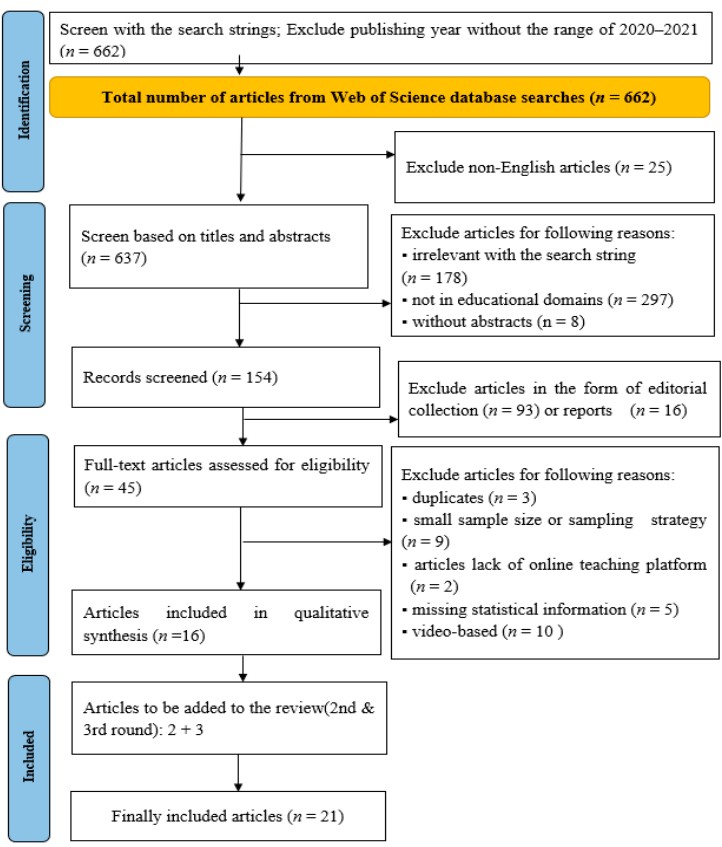

**Figure 1.** A PRISMA-based flow chart of the literature search.

### 3.3. Data Appraisal and Analysis

We used critical appraisal tools from the University of the West of England Framework [70,71]. We refined the results and scored each selected paper to get high-quality studies with STARLITE by removing the irrelevant and lower-quality results [71,72]. Shown in Appendices A and B

Two reviewers included and excluded the literature. If they could not reach an agreement on any decision, a third reviewer would be invited to finalize it. The inter-rater reliability reached a high level (k = 0.93). Inter-rater reliability used the formula suggested by Miles and Huberman [73]. We screened articles by the following steps using

the inclusion criteria listed in Figure 1. The articles were selected using the inclusion criteria shown in Table 1. Specifically, we implemented the inclusion and exclusion based on nine steps.

Step 1: We searched the Web of Science database in November 2021 and restricted the publishing year to 2020–2021, resulting in 662 articles.

Step 2: We excluded non-English articles, leading to 637 articles.

Step 3: We read the titles and abstracts (some literature had no abstracts) and excluded 483 articles that were irrelevant to the search string or did not belong to educational areas (education, computer, language, linguistics, literature, mathematic, music, chemistry, sociology, history, geography, biology, and physics, etc.).

Step 4: We filtered out editorial collections, reports, and review articles, leading to 45 articles.

Step 5: We excluded the duplicates and measured the sample size and sampling strategy. This resulted in 18 articles.

Step 6: We filtered articles short of online teaching platforms, leading to 16 articles.

Step 7: We used a non-systematic hand search after the first searching round to avoid missing additional articles. The hand search resulted in 2 articles that met the inclusion criteria.

Step 8: In December 2021, we conducted the third-round search using the original keywords in the Web of Science using the same criteria, and there was little difference. We screened the titles and abstracts, resulting in the addition of 3 articles.

Step 9: We used a STARLITE appraisal and analyzed the 21 articles.

**Table 1.** Data criteria and appraisal.

| Criteria (Points) Article | S (5) | T (5) | A (5) | R (2020–2021) (5) | L (5) | I (5) | T (5) | E (5) | Total Score (40) |
|---|---|---|---|---|---|---|---|---|---|
| Rodriguez-Segura et al. (2020) | 5 | 5 | 5 | 5 | 4 | 5 | 5 | 5 | 39 |
| Moorhouse (2021) | 2 | 2 | 3 | 5 | 3 | 3 | 5 | 4 | 27 |
| Collie (2021) | 4 | 5 | 4 | 5 | 4 | 4 | 5 | 5 | 27 |
| Lei & So (2021) | 5 | 5 | 5 | 5 | 5 | 4 | 4 | 5 | 38 |
| Hussein et al. (2020) | 3 | 5 | 4 | 5 | 4 | 4 | 5 | 5 | 35 |
| Truzoli et al. (2021) | 3 | 4 | 4 | 5 | 4 | 4 | 5 | 5 | 34 |
| Alves et al. (2021) | 4 | 5 | 4 | 5 | 5 | 4 | 5 | 5 | 37 |
| Casacchia et al. (2021) | 4 | 5 | 4 | 5 | 5 | 5 | 5 | 5 | 38 |
| Daumiller et al. (2021) | 4 | 5 | 4 | 5 | 5 | 4 | 5 | 5 | 37 |
| Aperribai et al. (2020) | 4 | 5 | 4 | 5 | 5 | 4 | 5 | 5 | 37 |
| Kraft et al. (2021) | 5 | 5 | 4 | 5 | 5 | 5 | 5 | 5 | 39 |
| Hong et al. (2021) | 4 | 5 | 4 | 5 | 4 | 4 | 5 | 5 | 36 |
| Masry-Herzalah & Dor-Haim (2021) | 4 | 5 | 3 | 5 | 5 | 5 | 5 | 4 | 36 |
| Konig, et al. (2020) | 5 | 5 | 4 | 5 | 5 | 5 | 5 | 5 | 39 |
| Perez-Calderon et al. (2021) | 3 | 4 | 4 | 5 | 4 | 4 | 5 | 5 | 34 |
| Tejedor et al. (2020) | 5 | 5 | 4 | 5 | 5 | 5 | 5 | 5 | 39 |
| Sanchez-Cruzado et al. (2021) | 5 | 5 | 5 | 5 | 5 | 5 | 5 | 5 | 40 |
| Almazova et al. (2020) | 4 | 5 | 5 | 5 | 4 | 4 | 5 | 5 | 37 |
| Sales et al. (2020) | 3 | 3 | 3 | 5 | 3 | 3 | 5 | 5 | 30 |
| Adov & Maeots (2021) | 3 | 3 | 3 | 5 | 3 | 4 | 5 | 5 | 31 |
| Perifanou et al. (2021) | 3 | 3 | 3 | 5 | 3 | 3 | 5 | 5 | 30 |

*3.4. Limitations to the Methods*

There are some limitations to the methods section. Firstly, the paper excluded articles written in a language other than English. We might miss some original articles, which might lead to publication bias. Secondly, we collected the data from the Web of Science Core Collection, the only database we could gain from our library. Furthermore, unpublished works, book chapters, and reports were excluded. We might miss some important articles which were included in other databases. It might bring us insufficient resources. Thirdly,

the paper mainly adopted a thematic analysis by using certain search criteria. We tried to improve the data collection procedures. However, no sample could be exhaustive because some samples were always eluding the search in the article selection process. There might be other potential biases in the selection of data given a thematic analysis.

## 4. Results

The COVID-19 pandemic has caused most schools to switch from conventional to online teaching worldwide. Many schools still use blended learning in some teaching processes, such as online discussion and submitting homework online. The lockdown or the COVID-19 pandemic revised the teacher's professional role, different from the classic understandings of teachers' work [74]. The results are presented according to the sequence of proposed research questions. The 21 publications are the main sources of citations in the results section of this article. These publications can be divided into three parts according to their contents. These articles mainly discuss teachers' professional role (about 25%), career satisfaction levels (about 30%), and teachers' digital literacy development (about 45%) during the COVID-19 pandemic. However, the results are not limited to these cited documents.

RQ1: What changes has teachers' professional role experienced during the COVID-19 pandemic worldwide?

### 4.1. Teacher's Professional Role

#### 4.1.1. Learning to Teach

Numerous studies (e.g., Rodriguez-Segura et al., Almazova et al., Kraft et al., Daumiller et al., Tejedor et al., Konig et al., Perifanou et al.) have shown that with the outbreak of the pandemic, the connotation of the teacher's professional role has undergone a more profound transformation and has become more complex [6,60,75–79]. With the pandemic of closed schools, teachers began to teach to meet higher quality requirements [80]. It was highlighted that teachers had to prepare for more profound teaching to make up for students' learning loss, resulting in a healing-informed teaching practice [81]. From in-person to online, high-achieving educational systems depend on the improved teacher professional role during the COVID-19 pandemic and future hybrid teaching. A survey from the United States showed that although before the outbreak of the COVID-19 pandemic, there had been a shortage of teachers, and on average less than 60% of teachers attached importance to new knowledge of cognition and instructional computer technology. This ratio reached 100% of teachers ranging from Sweden, Spain, Vietnam, Singapore, the Philippines, to China [82,83]. The role of teachers has been updated and reinforced at an unprecedented speed to meet the needs of today's society.

#### 4.1.2. Online Learning Guardian of Emotions

The world has witnessed a variation in professional teacher roles from the on-site teaching practice due to the COVID-19 pandemic [84]. More was to be conducted by a teacher-led model in online teaching and learning processes, which was different from the student-centered method. Teachers played a more crucial role in monitoring students' learning effects, and their psychological or technical problems appeared in online teaching platforms. Recent studies (e.g., Almazova et al., Daumiller et al., Sales et al., Adov & Maeots) showed that teachers needed to positively influence their students who had problems with self-regulated learning ability, attention, and computer literacy [60,76,84,85]. Furthermore, students were inclined to fall into depression or frustration when they met difficulties with distant learning. They could question their ability to learn. Teachers needed to guard learners to adapt and pay attention to their psychological acceptance [86].

RQ2: Are teachers satisfied with their teaching profession during the COVID-19 pandemic worldwide?

### 4.2. Teacher's Career Satisfaction

#### 4.2.1. Decline of Satisfaction Levels

Numerous studies (e.g., Almazova et al., Kraft et al., Daumille et al., Konig et al., Sales et al., Lei & So, Hussein et al., Alves et al., Truzoli et al., Aperribai et al., Hong et al.) revealed that teacher's satisfaction levels dropped and found that they were quickly in emotional exhaustion with the advent of the COVID-19 [60,75,76,78,84,87–92]. Although teaching half in-person and half distantly now could accelerate teacher's stress, which was closely linked with changes [93], teachers remained frustrated. They felt a sudden sense of failure and depression. Furthermore, teacher's satisfaction could be improved via convenient learning management systems [88]. Some anonymous online questionnaires regarding teacher's satisfaction levels carried out in some countries, such as Hong Kong [94], Portugal [89], and Iraq [88], revealed that teacher's professional satisfaction and well-being reduced. In contrast, their positive perception of satisfaction could be testified before the pandemic, which caused some stress and concern about teacher's future careers and teaching difficulties. Furthermore, research showed that teacher's mental conditions were impacted between September 2020 and October 2020 in Australia [95]. Overall, the COVID-19 pandemic dilapidated the foundation of teacher's emotional satisfaction, derived from a face-to-face student–teacher relationship in China, Japan, and the United States [90,96]. Available reports in a few studies pointed out that female teachers had survived lower satisfaction levels [97]. However, existing literature pointed out that teacher's satisfaction had no change during the pandemic and no significant predictors impacted teacher's satisfaction [98]. Numerous studies showed that teacher's satisfaction levels dropped with the COVID-19 outbreak although some factors were beyond control.

#### 4.2.2. Solutions

More research literature (e.g., Almazova et al., Kraft et al., Daumille et al., Konig et al., Sales et al., Lei & So, Hussein et al., Alves, et al., Truzoli et al., Aperribai, et al., Hong et al.) aimed to improve school teacher's satisfaction levels. Teacher's sense of satisfaction could be supported with working conditions [60,75,76,78,84,87–92]. Moreover, keeping their satisfaction could be critical to improving teacher's professional well-being [88]. Strong communities and school service quality could help promote teacher's satisfaction in the pandemic crisis [88]. If online teaching had become the primary method during and beyond the pandemic, teachers could appreciate this challenge as part of the education system, which was helpful to ease their dissatisfaction levels [97]. So, it was necessary to explore teacher's potential and competence towards the shift and motivate them to achieve teaching approach goals via training programs. Meanwhile, teacher's ability development could reduce their work reluctance and dissatisfaction [76]. Physical activities seemed to affect teacher's satisfaction levels. Data from the research showed that physical activity acted as a curb during the COVID-19 pandemic [91]. Still, some literature demonstrated that teacher's satisfaction levels rose if workloads, parenting stress, and work–family conflicts were reduced [92].

The COVID-19 pandemic has been affecting school teaching and learning processes for two years. Numerous studies have proved that the pandemic resulted in lower teacher's satisfaction levels. However, we could substantially adjust the current situation and make specific guidance interventions in the long run.

RQ3: What is the role of digital literacy during the COVID-19 pandemic worldwide?

### 4.3. Development of Teacher's Digital Literacy

#### 4.3.1. Educational Policy

Some studies (e.g., Almazova et al., Kraft et al., Daumille et al., Konig et al., Sales et al., Lei & So, Hussein et al., Alves, et al., Truzoli et al., Aperribai, et al., Hong et al.) revealed that the concept of teacher's digital literacy development has been universally accepted in many countries and expected to be blended correspondingly within school education development with the advent of the COVID-19 pandemic [60,75,76,78,84,87–92]. Many

changes adapted gradually to enhance the online educational environment in countries ranging from the United States, China, Spain, Italy, to Canada [99–101]. Local authorities provided some guidelines and encouraged new digital media, new digital competencies, new digital sources, and a new digital learning environment to facilitate sharing resources, which aimed at a multidimensional advancement of teacher's digital literacy [77]. Mainly, digital skills and knowledge were ensured in higher education. Almost all governments and countries in the world strived to improve teacher's digital literacy development at all costs. Organizational collaboration development was also emphasized in many governments worldwide, such as Italy, which built up digital platforms in schools. In Spain, governments provided numerous training programs, such as INTEF, to school staff purposefully and systematically [3].

### 4.3.2. Teacher Involvement

Current studies (e.g., Daumiller et al., Konig et al., Adov & Maeots, Masry-Herzalah & Dor-Haim) showed that digital literacy played a leading role in high-quality education in current situations [76,78,85,102]. Teachers at all academic levels were eager to develop their digital literacy in a short period owing to the virtual teaching and learning format [100]. They were compelled to adapt to online teaching. It seemed that digital competence had become a necessity in education [102,103]. Furthermore, teaching communities had urgently requested to gain digital skills for information collection, digital teaching content creation, communication, and collaboration. Still, teacher's feedback activities were favored to promote their feedback literacy [104]. However, teacher's digital competencies were low and had to be strengthened, especially in creating digital content, QR codes, and programs to modify software and apps [3].

RQ4: Can online higher education be an active agent for the change?

### 4.4. An Active Agent

Given that the pandemic continues to spread and derail normal operations, educational institutes adopted extensive online provisions through virtual learning platforms in some countries [1]. Online higher education is still the right choice for most students and is becoming increasingly popular as an attractive option for students during the COVID-19 pandemic. Students' dropout from online higher education could be high for many reasons and online higher education still had much room to enhance students' educational experience [1]. Furthermore, the COVID-19 pandemic helped online higher education gain market popularity and leverage, intentionally or unintentionally.

Overall, due to the paradigm change caused by the COVID-19 pandemic and the shift from traditional in-class education to online education, online higher education should be a promoter of development [1]. As an active agent, it is also expected to handle risks flexibly. It somehow succeeded in realizing more learning opportunities worldwide, which people had promised before. Moreover, in this new online higher education context, those with time, distance, and access difficulties were provided with genuinely accessible learning opportunities. Moreover, as a real change agent, online higher education yielded a substantial return to the investment for universities [1]. Even the pandemic has accelerated online higher education to become an active agent.

RQ5: Does online technology provide a sustainable education in terms of academic success and job readiness before and during COVID-19?

### 4.5. The Role of Online Technology

It was found that online-technology-based sustainable education might be the main tool for making money [69]. It could not ensure a sustainable education system for better educational development [105]. Online technology in the name of a sustainable education might mislead students for lack of a supply cycle and a specific demand [69]. Online-technology-based sustainable education was a tool to earn money. It would tarnish the reputation of higher education for a long time. Consequently, the role of sustainable

education will be doubtful [72]. Online technology would aggravate the diploma crisis during the COVID-19 pandemic [72]. So, a rigid and regulatory framework was needed to ensure the quality of sustainable education; the framework should be driven with application and required revision continuously during the COVID-19 pandemic. Some literature explored a better technique that might be employed to ensure a sustainable education between industry and universities via relieving the diploma disease crisis [64].

Some studies pointed out that there were both pros and cons of online technology [2]. Students in online education had higher grades than they did in the conventional face-to-face teaching method. Online technology might help improve students' academic performance, although students' academic achievements were doubtful, as there was a possibility to fabricate grading [64]. Online technology provided a lot of innovative ideas in many cases to improve sustainable education, yet pieces of evidence suggested that students performed well in job readiness before the COVID-19 pandemic [2]. Furthermore, online technology had shortcomings in providing sustainable education during the COVID-19 pandemic [2]. It might threaten sustainable education in some developing countries [2]. The efficacy of online knowledge and skills delivery raised doubts, as students with better grades did not correlate with job readiness and real competencies. Although the technology seemed to meet the requirements of sustainable education, it damaged its reputation and essential nature [2]. Graduates' academic success might not reflect the realities of the job market. Students' credentials or qualifications did not completely indicate their job readiness.

## 5. Discussion

Almost all teaching staff coped with the sudden shift from the face-to-face teaching model to online teaching during the COVID-19 pandemic. They had to struggle to adapt to the additional stresses and workloads [55]. Accordingly, this caused changes in teacher's professional role, decline in their satisfaction levels, and many digital literacy challenges, because teachers had to balance their responsibilities, teaching, work-life, and development during the urgent imperative to move to online teaching. Teachers had more obligations, which often entailed teachers facing greater psychological pressure. In addition, pressure could also result from a lack of technical support. There are still teachers with no or little digital literacy in online teaching, as they come from all ages and backgrounds.

The present study mainly discussed teacher's role, career satisfaction, and digital literacy at the outbreak of the COVID-19 pandemic. The study found that improving teacher's career satisfaction helped deepen the recognition of teacher's roles and digital literacy. Furthermore, recognizing the teacher's role promoted teacher's career satisfaction and improved teacher's digital literacy aspirations. At the same time, the development of teacher's digital literacy helps to interpret teacher's roles better and increase their satisfaction. The three aspects (Figure 2), mutually intertwined, could provide a high-quality education [85]. Furthermore, teacher's full-scale development will continue in the post-COVID-19 epidemic.

However, each of the three aspects is different from the other two, and there are divergent ways to realize them. Teacher's digital literacy development relied on academic environment support, faculty members' readiness, and students' readiness. It is vitally important to maximize the positive roles of teachers in the education process. Teachers' satisfaction levels could be elevated by psychological, methodological, and technological support. However, the COVID-19 pandemic is not over, and there is still a long way to go as the potential of online teaching continues. We could weed out hindering factors to ensure the comprehensive development of educators.

The study also found that online higher education presents a series of advantages, such as the delivery of knowledge and skills, application of techniques, and timely information management due to the transition period [5]. Online higher education applied timely educational applications to meet the community needs [5]. Online higher education uses the platform to make teachers and students exchange information. Furthermore, it produces

a good environment for all participants [6]. The implementation of online higher education facilitated the teaching and learning process. In addition, students' cognitive abilities develop better, and they are active participants in the process. Thus, their academic performance gets improved.

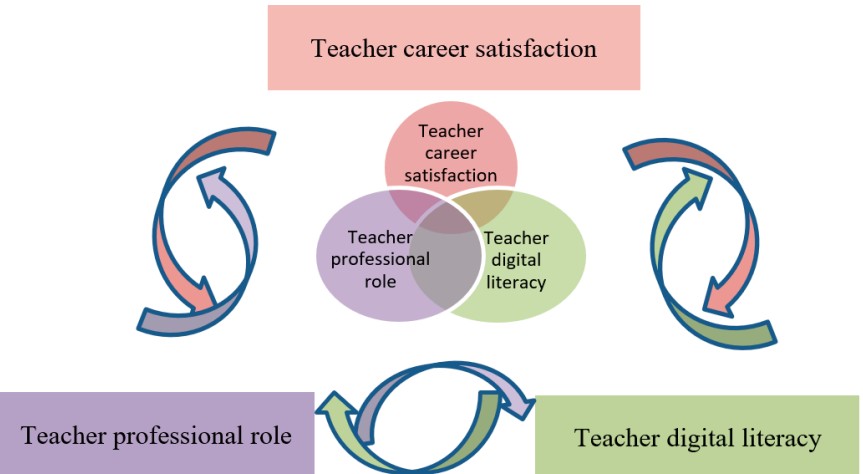

**Figure 2.** Intertwined relationships among variables identified.

Online technology in providing sustainable education should not be a permanent option in students' academic success and job readiness [5]. Online technology is a flexible and convenient online learning way during the COVID-19 pandemic. It contributes to the delivery of knowledge and skills compared with the conventional learning model. However, strict and prudent measures are also required to ensure its benefits [2]. Graduates' academic performance may measure the effectiveness of a sustainable education [67]. However, there are no reciprocal connections between academic success and job readiness. Academic performance is only the main parameter to measure a sustainable educational effect [4]. Somehow, the COVID-19 pandemic makes academic achievement overvalued. An online-technology-based sustainable education of academic success and job readiness during the COVID-19 pandemic becomes dysfunctional. Overall, we should explore a comprehensive approach to a sustainable education via online technology during COVID-19.

## 6. Conclusions

### 6.1. Major Findings

The study found that teachers' professional roles changed complicatedly and that teachers were assigned more tasks in the online teaching process. Along with the change in the educational environment, improving teacher's digital literacy becomes urgent during the pandemic. Overall, from the standpoint of educational administrators, they may be harnessed to support teachers based on the correlations of the three factors during the COVID-19 pandemic or in the future, as these factors showed some difficulties and constraints that teachers confronted in the online teaching-learning process.

The study also attempted to explore some common perceptions regarding the role of online higher education and online technology. We have focused on the actual change and status of online higher education. We found that online higher education could be an active agent for the changes, and online technology might partly provide a sustainable education in terms of academic success and job readiness before and during COVID-19. Generally, online higher education and technology benefited the society. Meanwhile, both could improve teachers' and students' perceptions. The COVID-19 pandemic severely damaged the economy worldwide, where online higher education could function to a certain degree.

*6.2. Limitations*

There are some limitations to the present study. First, it was not easy to precisely measure the effectiveness of online education. Second, we tried to reveal the changes in teacher's role, satisfaction levels, and digital literacy development. Due to the length of the teaching content and students' online learning behavior in the online teaching-learning process, it was complicated to accurately investigate the changes in teacher's professional role, career satisfaction levels, and the real need for their digital literacy during the COVID-19 pandemic. Third, it was necessary to provide sufficient evidence to support the study. It was hard to gain sufficient support due to the inadequate resources. In addition, we could not exhaust all samples due to the limited library sources. So the reliability of results might be lowered. Moreover, the results of various studies are inconsistent. Some empirical studies concluded that the COVID-19 pandemic enhanced teacher's professional role and satisfaction levels, and that teacher's digital literacy was well developed, while others revealed the reverse results.

Although the present study tried to demonstrate some perceptions regarding the role of online higher education and online technology, we admit that it is difficult to know to what extent online higher education as an active agent is realized during the pandemic. Moreover, its impact on higher education still needs further demonstration. In addition, we need to further explore whether online technology provides a sustainable education in terms of academic success and job readiness before and during the COVID-19 pandemic. The present study adopts a qualitative method which could be enhanced by quantitative methods.

*6.3. Future Research Directions*

The present study focused on teacher's role, satisfaction, and digital literacy. Future research could explore the factors that hinder them through empirical experiments, for the pandemic will most likely continue for some time. Future research could include more factors that might influence the educational outcomes during the pandemic. Future research could explore teacher's intrinsic cognitive loads owing to paradoxical findings. Another research stream is to measure teacher satisfaction levels and investigate what policymakers could do in high-quality educator preparation. We might take the next plausible step to clearly answer these research questions via convincing methods. Future research might also carry out more case studies via representative samples and enough data.

**Author Contributions:** Conceptualization, M.L. and Z.Y.; methodology, M.L.; software, M.L.; validation, M.L. and Z.Y.; formal analysis, M.L. and Z.Y.; investigation, M.L. and Z.Y.; resources, M.L.; data curation, M.L.; writing—original draft preparation, M.L.; writing—review and editing, Z.Y.; visualization, M.L.; supervision, Z.Y.; project administration, Z.Y.; funding acquisition, Z.Y. All authors have read and agreed to the published version of the manuscript.

**Funding:** This research was funded by the "2019 MOOC of Beijing Language and Culture University (Important)", grant number "MOOC201902", the "Special fund of Beijing Co-construction Project-Research and reform of the "Undergraduate Teaching Reform and Innovation Project" of Beijing higher education in 2020-innovative "multilingual +" excellent talent training system", grant number "202010032003", the "The research project of Graduate Students of Beijing Language and Culture University 'Xi Jinping: The Governance of China'", grant number "SJTS202108", and the " 'Introduction to Linguistics' of online and offline mixed courses in Beijing Language and Culture University in 2020".

**Institutional Review Board Statement:** Ethical review and approval were waived for this study because this study does not involve humans or animals.

**Informed Consent Statement:** Not applicable.

**Data Availability Statement:** Not applicable.

**Conflicts of Interest:** The authors declare no conflict of interest.

## Appendix A

**Table A1.** University of West England Framework for Critically Appraising Research Articles [70,71].

| Section and Topic | Checklist Item (s) |
|---|---|
| **Introduction** | |
| Rationale and objectives | A clear rationale and research questions or objectives |
| **Methods** | |
| Search strategy | Appropriate methods used to research purposes |
| Selection process | reasonable research design |
| | the number of participants |
| | risks of biased process |
| **Data collection process** | |
| Data items | A explicit description on the data collected |
| | A clear statement on the data analysis |
| Assessment methods | Quantitative, qualitative, or mixed methods |
| **Ethics** | |
| criterion | A clear statement about gaining consent |
| | A statement maintaining data anonymity and confidentiality |
| **Results** | |
| Result reliability | The results related back to the literature review |
| | Availability of data |
| | A clear presentation of results |
| | Any possible limitations of results |
| | Multiple presentation methods |
| Conclusions | A statement of further research direction(s) |
| | Any further research suggestions |

## Appendix B

**Table A2.** The STARLITE Evaluation [71,72].

| Component | Explanation |
|---|---|
| S: Sampling strategy | Comprehensive: the sample should be comprehensive |
| | enough to be representative. |
| | Selective: the sample should be scientifically selected. |
| | Purposive: the sample should source from related fields. |
| T: Type of studies | Fully reported: the sample should clearly |
| | explain the specific study type. |
| | Partially reported: the sample sometimes |
| | generally describes the study type. |

**Table A2.** *Cont.*

| Component | Explanation |
|---|---|
| A: Approaches | Approaches could retrieve literature from online databases and |
| | directly search them online. |
| R: Range of years (start date- end date) | The sample should source from a certain period. |
| L: Limits | There are some limits on sampling such as the language |
| | used and research methods adopted. |
| I: Inclusion and exclusions | There are criteria to include or exclude literature. |
| T: Terms used | There must be terms to retrieve high-quality literature. |
| E: Electronic sources | Samples may be from online databases, free publications or |
| | other electronic sources. |

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
