# Peer review of "Teachers’ Satisfaction, Role, and Digital Literacy during the COVID-19 Pandemic"

_sustainability, doi:10.3390/su14031121_

Round 1
Reviewer 1 Report
Abstract: The last two sentences of the abstract need to be clarified. Namely, what is the potential impact of the results of the author’s study?
Introduction: The introductory section should illustrate more comprehensively the significance of the study. Perhaps, it may even give a general idea about the research question(s) that the study attempts to answer.
Methodology: The method section should include details of the specific method used to analyze the data. If each of the three research questions drives the author's analyses, how is the extraction of data performed? Is thematic analysis considered? How were potential biases in the selection of information avoided once the articles were selected? Is there a measure of inter-rater reliability? Without clear and detailed information about the data extraction procedure used by the author is difficult to ascertain the results of the study.
Line 178-179. Please clarify the following sentence: “Twenty-one publications were coded as depicted to answer the research questions in 179 Table 1, as the primary citations were included to be suitable for inclusion.”
The grammatical structure and word choices of the entire manuscript need to be carefully examined.
Author Response
Response to reviewer comments
Dear reviewer,
Thank you very much for your careful review work. We have conducted substantial revisions based on your constructive suggestions as follows.
Please feel free to contact us if you have any other questions.
Best regards!
Sincerely yours,
Dr. Li Ming
Prof., Dr. Yu Zhonggen
Beijing Language and Culture University
Response to Reviewer 1:
(1) Abstract: The last two sentences of the abstract need to be clarified. Namely, what is the potential impact of the results of the author’s study?
Response:Thank you very much for your positive comments and constructive suggestions. According to your advice, we have improved the clarity of the last two sentences in the revised manuscript.
Here are the details:
“After the COVID-19 pandemic, it is necessary to conduct a blended-teaching model in educational institutes. Teachers should have adequate digital literacy to meet the new needs of the currently innovative educational model in the future. In addition, the study reveals that teacher digital literacy level, career satisfaction, and professional role are significantly correlated. We will measure to what degree the three factors affect the online teaching and learning process. Ultimately, the study may provide some suggestions for methodological and educational strategies.”
(2) Introduction: The introductory section should illustrate more comprehensively the significance of the study. Perhaps, it may even give a general idea about the research question(s) that the study attempts to answer.
Response:Thank you for your thoughtful suggestions. According to your advice, we conducted some supplementary on the significance of the study to support our research. To more clearly present the background of the study, we have revised the introduction section. All additions have been added in the revised manuscript. Please see the introduction section.
Here are the details:
“Teachers must have adequate digital literacy to teach online, which is required in the current educational model. Nevertheless, digital literacy alone could not favor the teaching process (Sanchez-Cruzado et al., 2021). Successful teaching process also involves teacher professional role and satisfaction. The three elements are intertwined and essential for overall online teaching and learning (Jalongo, 2021). The study explores the changes in teacher professional role and career satisfaction levels and the challenges of their digital literacy during the COVID-19 pandemic worldwide. The combination of three elements may enable teachers to perform duties in schools better. Moreover, the study explores the changes of the role of online higher education as an active agent during the COVID-19 pandemic because teachers and students are mainly dependent on online technology-based platforms to sustain education.”
References:
- Jalongo, M. R. (2021). The effects of COVID-19 on early childhood education and care: Research and resources for children, families, teachers, and teacher educators. Early Childhood Education Journal, 49, 763-774. Https://doi.org/10.1007/s10643-021-01208-y
- Sanchez-Cruzado, C., Campion, R. S., & Sanchez-Compa, M. T. (2021). Teacher digital literacy: The indisputable challenge after COVID-19. Sustainability, 13. Https://doi.org/10.3390/su13041858
(3) Methodology: The method section should include details of the specific method used to analyze the data. If each of the three research questions drives the author's analyses, how is the extraction of data performed? Is thematic analysis considered? How were potential biases in the selection of information avoided once the articles were selected? Is there a measure of inter-rater reliability? Without clear and detailed information about the data extraction procedure used by the author is difficult to ascertain the results of the study.
Response: Thank you very much for your thoughtful comments. We quite agree with you. We have realized that procedures for selection and evaluation are important. We collected a large amount of data from the Web of Science Core Collection, the only database we can gain from our library. This study collected data for a specific time point (i.e., the COVID-19 outbreak). Moreover, this study investigates the changes in teachers’ professional role and career satisfaction levels and the challenges of their digital literacy that they met during the COVID-19 pandemic worldwide. So this paper is only a qualitative study and not a quantitative study. We have used critical appraisal tools from The University of West England Framework to analyze (Moule et al., 2021; Yu et al., 2020). Data analyses were performed using STARLITE (Booth, 2006; Yu et al., 2020). Also, specific details of the scales are not shown in the manuscript due to space reasons. Furthermore, we have listed the overall checklist items that some of the studies that used the filter criteria (Table 1). We have also displayed the main scoring criteria (Table 2).
According to your advice, we have revised the method section and added more detailed procedures to extract data performed. We have realized there may have potential biases in selecting information given thematic analysis, and some potential biases were unavoidable in the article selection process. So we have added some limitations and inter-rater reliability part in the paper. All additions in the revised manuscript have been marked.
Here are the details:
“3.2 Main inclusion criteria
We firstly screened the articles mainly based on the appropriateness of abstracts. Secondly, we included the articles if they fulfilled the following conditions: 1) The publications were made available in the form of articles; 2) The empirical study demonstrated teachers’ satisfaction, role, and digital literacy during the COVID-19 pandemic; 3) The articles focused on educational domains; 4) The studies were written in the English language; 5) The dependent variables should be teachers’ satisfaction, role, or digital literacy; 6) The studies provided sufficient data and analyses from which the conclusions were drawn (standard deviations, F, or means were basically required). Based on the inclusion criteria, 21 articles were included in this study (Figure 1).
3.3. Data appraisal and analysis
We use critical appraisal tools from The University of West England Framework [1, 2]. We refined the results and scored each selected paper to get high-quality studies with STARLITE by removing the irrelevant and lower-quality results (Booth, 2006; Yu et al., 2020).
Two reviewers included and excluded the literature. If they could not reach an agreement on any decision, a third reviewer would be invited to finalize it. The inter-rater reliability reaches a higher level (k = 0.93). Inter-rater reliability has used the formula suggested by Miles and Huberman (Miles & Huberman, 1994). We screened articles in the following steps using the inclusion criteria listed in Figure 1. The articles were selected with the inclusion criteria shown in Table 1. Specifically, we implemented the inclusion and exclusion based on nine steps.
Step 1: We searched the Web of Science database in November 2021 and restricted the publishing year to 2020-2021, resulting in 662 articles.
Step 2: We excluded non-English articles, leading to 637 articles.
Step 3: We read the titles and abstracts (some literature had no abstracts) and ex-cluded 483 articles that were irrelevant to the search string or didn’t belong to educational areas (education, computer, language, linguistics, literature, mathematic, music, chemistry, sociology, history, geography, biology, and physics, etc.).
Step 4: We filtered an editorial collection, reports, or review articles, leading to 45 articles.
Step 5: We excluded the duplicates and measured the sample size and sampling strategy. It resulted in 18 articles.
Step 6: We filtered articles short of online teaching platforms, leading to 16 articles.
Step 7: We used a non-systematic hand search after the first searching round to avoid missing additional articles. The hand search resulted in 2 articles that met the inclusion criteria.
Step 8: In December 2021, we conducted the third round search using the original keyword across the Web of Science with the same criteria, and there was little difference. We screened the titles and abstracts, resulting in the addition of 3 articles.
Step 9: We used a STARLITE appraisal and analyzed the twenty-one articles.
3.4 Limitations to the methods
“There are some limitations to the method section. Firstly, the paper excluded articles written in a language other than English. We might miss some original articles, which might lead to publication bias. Secondly, we collected the data from the Web of Science Core Collection, the only database we could gain from our library. Furthermore, un-published works, book chapters, or reports were excluded. We might miss some important articles which were included in other databases. It might bring us insufficient resources. Thirdly, the paper mainly adopted a thematic analysis by using certain search criteria. We tried to improve the data collection procedures. However, no sample could be exhaustive because some samples were always eluding the search in the article selection process. There might be other potential biases in the selection of data given a thematic analysis.”
Table 1. Detailed selection procedures
Section and Topic |
Item |
Checklist item |
TITLE |
||
Title |
1 |
Identify the report as a systematic review. |
ABSTRACT |
||
Abstract |
2 |
See the PRISMA 2020 for Abstracts checklist. |
INTRODUCTION |
||
Rationale |
3 |
Describe the rationale for the review in the context of existing knowledge. |
Objectives |
4 |
Provide an explicit statement of the objective(s) or question(s) the review addresses. |
METHODS |
||
Eligibility criteria |
5 |
Specify the inclusion and exclusion criteria for the review and how studies were grouped for the syntheses. |
Information sources |
6 |
Specify all databases, registers, websites, organisations, reference lists and other sources searched or consulted to identify studies. Specify the date when each source was last searched or consulted. |
Search strategy |
7 |
Present the full search strategies for all databases, registers and websites, including any filters and limits used. |
Selection process |
8 |
Specify the methods used to decide whether a study met the inclusion criteria of the review, including how many reviewers screened each record and each report retrieved, whether they worked independently, and if applicable, details of automation tools used in the process. |
Data collection process |
9 |
Specify the methods used to collect data from reports, including how many reviewers collected data from each report, whether they worked independently, any processes for obtaining or confirming data from study investigators, and if applicable, details of automation tools used in the process. |
Data items |
10a |
List and define all outcomes for which data were sought. Specify whether all results that were compatible with each outcome domain in each study were sought (e.g. for all measures, time points, analyses), and if not, the methods used to decide which results to collect. |
10b |
List and define all other variables for which data were sought (e.g. participant and intervention characteristics, funding sources). Describe any assumptions made about any missing or unclear information. |
|
Study risk of bias assessment |
11 |
Specify the methods used to assess risk of bias in the included studies, including details of the tool(s) used, how many reviewers assessed each study and whether they worked independently, and if applicable, details of automation tools used in the process. |
Effect measures |
12 |
Specify for each outcome the effect measure(s) (e.g. risk ratio, mean difference) used in the synthesis or presentation of results. |
Synthesis methods |
13a |
Describe the processes used to decide which studies were eligible for each synthesis (e.g. tabulating the study intervention characteristics and comparing against the planned groups for each synthesis (item #5)). |
13b |
Describe any methods required to prepare the data for presentation or synthesis, such as handling of missing summary statistics, or data conversions. |
|
13c |
Describe any methods used to tabulate or visually display results of individual studies and syntheses. |
|
13d |
Describe any methods used to synthesize results and provide a rationale for the choice(s). If meta-analysis was performed, describe the model(s), method(s) to identify the presence and extent of statistical heterogeneity, and software package(s) used. |
|
13e |
Describe any methods used to explore possible causes of heterogeneity among study results (e.g. subgroup analysis, meta-regression). |
|
13f |
Describe any sensitivity analyses conducted to assess robustness of the synthesized results. |
|
Reporting bias assessment |
14 |
Describe any methods used to assess risk of bias due to missing results in a synthesis (arising from reporting biases). |
Certainty assessment |
15 |
Describe any methods used to assess certainty (or confidence) in the body of evidence for an outcome. |
RESULTS |
||
Study selection |
16a |
Describe the results of the search and selection process, from the number of records identified in the search to the number of studies included in the review, ideally using a flow diagram. |
16b |
Cite studies that might appear to meet the inclusion criteria, but which were excluded, and explain why they were excluded. |
|
Study characteristics |
17 |
Cite each included study and present its characteristics. |
Risk of bias in studies |
18 |
Present assessments of risk of bias for each included study. |
Results of individual studies |
19 |
For all outcomes, present, for each study: (a) summary statistics for each group (where appropriate) and (b) an effect estimate and its precision (e.g. confidence/credible interval), ideally using structured tables or plots. |
Results of syntheses |
20a |
For each synthesis, briefly summarise the characteristics and risk of bias among contributing studies. |
20b |
Present results of all statistical syntheses conducted. If meta-analysis was done, present for each the summary estimate and its precision (e.g. confidence/credible interval) and measures of statistical heterogeneity. If comparing groups, describe the direction of the effect. |
|
20c |
Present results of all investigations of possible causes of heterogeneity among study results. |
|
20d |
Present results of all sensitivity analyses conducted to assess the robustness of the synthesized results. |
|
Reporting biases |
21 |
Present assessments of risk of bias due to missing results (arising from reporting biases) for each synthesis assessed. |
Certainty of evidence |
22 |
Present assessments of certainty (or confidence) in the body of evidence for each outcome assessed. |
DISCUSSION |
||
Discussion |
23a |
Provide a general interpretation of the results in the context of other evidence. |
23b |
Discuss any limitations of the evidence included in the review. |
|
23c |
Discuss any limitations of the review processes used. |
|
23d |
Discuss implications of the results for practice, policy, and future research. |
|
OTHER INFORMATION |
||
Registration and protocol |
24a |
Provide registration information for the review, including register name and registration number, or state that the review was not registered. |
24b |
Indicate where the review protocol can be accessed, or state that a protocol was not prepared. |
|
24c |
Describe and explain any amendments to information provided at registration or in the protocol. |
|
|
|
|
Support |
25 |
Describe sources of financial or non-financial support for the review, and the role of the funders or sponsors in the review. |
Competing interests |
26 |
Declare any competing interests of review authors. |
Availability of data, code and other materials |
27 |
Report which of the following are publicly available and where they can be found: template data collection forms; data extracted from included studies; data used for all analyses; analytic code; any other materials used in the review. |
Table 2. The STARLITE Evaluation
Component |
Explanation |
S: Sampling strategy |
Comprehensive: the sample should be comprehensive enough to be representative. Selective: the sample should be scientifically selected. Purposive: the sample should source from related fields. |
T: Type of studies |
Fully reported: the sample should clearly explain the specific study type. Partially reported: the sample sometimes generally describes the study type. |
A: Approaches |
Approaches could retrieve literature from online databases and directly search them online. |
L: Limits |
There are some limits on sampling such as the language used and research methods adopted. |
T: Terms used |
There must be terms to retrieve high-quality literature. |
References:
- Booth, A. (2006). “Brimful of STARLITE” toward standards for reporting literature searches. Journal of the Medical Library Association, 94, 421–429.
- Miles, M. B., & Huberman, A. M. (1994). Qualitative data analysis: An expanded
Sourcebook (2nd ed.). Thousand Oaks: Sage Publications.
- Moule, P., Pontin, D., Gilchrist, M., & Ingram, R. Critical appraisal framework. Available online: Http://learntech.uwe.ac.uk/da/Default.aspx?pageid=1445 (accessed on 22 December 2021)
- Yu, Z. G., Gao, M. L., & Wang, L. F. (2020). The effect of educational games on learning outcomes, student motivation, engagement and satisfaction. Journal of Educational Computing Research, 59, 522-546. Https://doi.org/10.1177/0735633120969214
(4) Line 178-179. Please clarify the following sentence: “Twenty-one publications were coded as depicted to answer the research questions in 179 Table 1, as the primary citations were included to be suitable for inclusion.”
Response: Thank you very much for your suggestion. Two reviewers included and excluded the literature. If they could reach an agreement on any decision, a third reviewer would be invited to make a decision. The inter-rater reliability reaches a higher level (k = 0.93).The Twenty-one publications are the main sources of citations for the results and discussion sections of the research. However, the result and discussion sections are not limited to these documents. These articles can be divided into three parts according to their contents. These articles mainly discuss teachers’ professional role, career satisfaction levels, and teachers’ digital literacy development during the COVID-19 pandemic. However, as we discussed in the paper, each article included more than one variable and interweaved three or more variables. On the other hand, we selected the twenty-one articles from the Web of Science Core Collection to conclude as objectively as possible.
(5) The grammatical structure and word choices of the entire manuscript need to be carefully examined.
Response: Thank you very much for your positive suggestions, and we carefully revised our paper according to your suggestions.

Reviewer 2 Report
Journal name: Sustainability (ISSN: 2071-1050)
Title: A Review of Teachers’ Satisfaction, Role, and Digital Literacy 2 during the COVID-19 Pandemic
Manuscript ID Sustainability 1540174
Thank you for inviting me to review this work. First of all, I would like to thank and acknowledge the effort and work done by the authors on this study.
In order to follow and understand the comments made on your work, I inform you that I will respect the order and structure of your manuscript
As for the abstract, I will tell you that it should provide basic information on the procedure and methodology used. You should briefly describe the techniques and instruments used and the main results. Briefly describe the sample. You leave an open question in the abstract when you refer in the last line to: The subsequent study will conduct an empirical experiment to testify the level of teacher satisfaction in China. You should clarify this question because it is confusing and does not specify whether it is part of the study or not.
Your study is a systematic review, you should indicate this in the abstract and describe the number of papers found and the final number of papers analyzed indicating the main and most important results.
To initiate a systematic review it is necessary to identify and convert the problem, uncertainty, or "knowledge gap" into a question that can be answered. Formulating a question means reducing it to clear and precise terms, identifying its main components. The PICO mnemonic can help you structure the four components of a clinical question. The relevant patients or population groups, the intervention (exposure or diagnostic procedure) of interest, as well as against whom the intervention is being compared and the appropriate (outcomes). I recommend that you justify whether you have used this technique, which is the one I recommend, or whether you have used another.
Regarding the database search. Firstly, the eligibility criteria should be defined according to the PICO approach, it does not specify the type of studies included and the exclusion criteria should also be specified. In the search strategy the authors indicate that the search has been performed in WOS journals, however, you do not say which databases were used or how this search was carried out. Remember to make the research methodology very clear so that your study can be replicated.
To ensure that a systematic review is valuable, authors should prepare a transparent, complete, and accurate account of why the review was done, what they did, and what they found. The PRISMA statement (2020) provides updated guidance on the submission of systematic reviews that reflect advances in methods for identifying, selecting, appraising, and synthesizing studies. I recommend that if you have used this methodology you make it explicit in your work.
From line 163 to 165 the authors make a very ambiguous statement; The results involved 77 articles. Moreover, manual screening removed and added some papers; 21 articles met the inclusion criteria. Please clarify that it means that with manual screening you removed and added some articles. Under what criteria did you do this.
The authors say; We use critical appraisal tools from The University of West England Framework [1, 2] (Appendix A). We removed the irrelevant and lower-quality results to refine the results and scored each selected paper to get high-quality studies with STARLITE [2, 3] (Appendix B). These elimination criteria should be clarified
In general, the inclusion and exclusion criteria are not sufficiently specified and the comments on these criteria are ambiguous and unclear.
In the results section, you should group the studies together and specifically cite the studies that provide evidence for the results you are commenting on. Do not use expressions such as some studies... refer to the specific studies that support your assertions.
Between lines 174 and 177 you state: We use critical appraisal tools from The University of West England Framework [1, 2] (Appendix A). We removed the irrelevant and lower-quality results to refine the results 175 and scored each selected paper to get high-quality studies with STARLITE [2, 3] (Appendix B).
How was the decision made to eliminate documents... how did the researcher consider the relevance or irrelevance of the studies... was the decision triangulated with the opinion of other researchers? Failure to adequately clarify this question could be considered subjective judgment on the part of the investigator.
The source selection scheme is not clear enough, the application of the criteria should result in the final 21 selected articles. Adjust the scheme so that the final result of checking the criteria and reading the abstract of the final 21 articles.
In the results, you should also include the percentages of the number of articles that address each of the research topics.
The discussion is not a true discussion, you do not analyze the results in light of the theoretical framework of your study. You should discuss the results. Please address the discussion in depth.
Conclusions should be rephrased, conclusions are not really presented. You repeat the research questions again, this is unnecessary. You should draw at least one or two conclusions from the dimensions that are analyzed in the results.
The section on limitations and future lines of research are not conclusions. Please extract this section from the conclusions.
Your paper requires a thorough review.
Author Response
Response to reviewer comments
Dear reviewer,
Thank you very much for your careful review work. We have conducted substantial revisions based on your constructive suggestions as follows.
Please feel free to contact us if you have any other questions.
Best regards!
Sincerely yours,
Dr. Li Ming
Prof., Dr. Yu Zhonggen
Beijing Language and Culture University
Response to Reviewer 2
It is a well-developed and pertinent study in the face of the main global emergency. It is a relevant work at this moment. However, it needs some improvements before publishing.
The subject of the study on satisfaction, the role and digital literacy of the teacher are well developed, but it would be good to clarify these concepts from the introduction for better clarity.
I like to read the conceptual framework; however, I like to see the link between the role of education and the interference of virtual learning in this role before outlining the goal, the objectives, and the research questions. Also, they must be aligned.
Response: Thank you very much for your suggestion. According to your advice, we have revised the manuscript's introduction section to illustrate teachers' satisfaction, professional role, and digital literacy more clearly. And we conducted a series of supplementary of the role of education and the interference of virtual learning to support our research. Consequently, we have added some literature in the revised manuscript. All additions in the manuscript have been marked. (Please see the introduction section in the revised manuscript).
Here are the details:
As frontline providers of education, teachers are increasingly important in educational settings, especially in the virtual teaching and learning environments. The COVID-19 pandemic accelerated the process of educational virtualization (Zamora-Antunano et al., 2021). It seems that education at all levels develops online virtual learning platforms (Zamora-Antunano et al., 2021). Teachers’ profession can be regarded as a motivator to use virtualization in teaching, where many different aspects in the teaching process are connected. The teacher professional role, as a pedagogue, can solve students’ problems (Rodriguez-Segura et al., 2020). Teacher career satisfaction is a pleasant mental state arising from their appreciation of work or experience (Rodriguez-Segura et al., 2020). It is important for teachers to feel satisfied with their work or profession. Teachers’ digital literacy indicates the ability to use digital resources and virtual learning platforms in the educational environment. Teachers equipped with basic digital literacy will be highly competitive in future online or classroom practice (Zamora-Antuñano et al., 2022). Overall, as a knowledge transmitter, the teacher plays a significant role in virtual educational settings during the COVID-19 pandemic.
References:
- Rodriguez-Segura, L., Zamora-Antunano, M. A., Rodriguez-Resendiz, J., Altamirano-Corro, J. A., Paredes-Garcia, W. J., & Cruz-Perez, M. A. (2020). Teaching challenges in COVID-19 scenery: Teams platform-based student satisfaction Approach, Sustainability, 12(18). Https://doi.org/10.3390/su12187514
- Zamora-Antuñano, M. A., Rodríguez-Reséndiz, J., Cruz-Pérez, M. A., Reséndiz Reséndiz, H., Paredes-García, W. J., & Díaz, J. A. G. (2022). Teachers’ perception in selecting virtual learning platforms: A case of Mexican highereducation during the COVID-19 crisis. Sustainability, 14(195). Https://doi.org/10.3390/su14010195
- Zamora-Antunano, M. A., Rodriguez-Resendiz, J., Segura, L. R., Perez, M. A. C., Corro, J. A. A., Paredes-Garcia, W. J., & Rodriguez-Resendiz, H. (2021). Analysis of emergency remote education in COVID-19 crisis focused on the perception of the teachers. Sustainability, 13(7). Https://doi.org/10.3390/su13073820
The author can get use the sources at ‘sustainable education for society the emergency: comparison between technological forecasting and social change before and during COVID-19’. I consider the discussion as the main opportunity of the article if it includes the objectives in the introduction, and integrates the questions:
Can online higher education be an active agent for change?
Does online technology provide a sustainable education comparison of academic success and job readiness before and during COVID-19?
Therefore, the author or authors should review the most recent articles to improve their research
- https://doi.org/10.3390/su14010195
- https://doi.org/10.3390/su13073820
- https://doi.org/10.3390/su12187514
It would be good to strengthen some arguments in the results section.
Response: Thank you very much for your positive comments and constructive suggestions. According to your advice, we conducted a series of supplementary to support our research. We have added the two research questions and some literature you suggest to this study. All additions have been added to the revised manuscript.
Here are the details:
Introduction
1.3
The year 2020 has witnessed the urgent need for online higher education, which was courageous in redesigning education fundamentals (Zamora-Antuñano et al., 2022). Numerous schools made prompt arrangements to full online provision. The implications of online higher education have gained significant prominence. Online higher education aims to get educators into the Internet space, expanding students’ learning opportunities (Alam, 2021). When the online education market rapidly developed within a few decades, higher education institutions adapted for online courses offering and design. Higher education with online technology may be a popular mode worldwide. Meanwhile, online technology drew numerous participants to use the online learning opportunities. These have triggered us to consider online higher education agents (Zamora-Antuñano et al., 2022).
Literature review
2.4. Online higher education
Online higher education refers to a revolutionary solution to different and unequal educational issues (Zamora-Antuñano et al., 2022). The adoption of online higher education would become beneficial for university education. Furthermore, higher education should fully take the form of online delivery (Zamora-Antuñano et al., 2022). At the early stage of online higher education, it aimed to offer non-traditional students access to educational opportunities. Current studies had acknowledged that online higher education became increasingly complicated and diverse in practice (Lee, 2017), especially when it encompassed traditional face-to-face teaching and learning formats and technology-based online education. Some studies revealed that online higher education was an active agent for it was interactive and collaborative in that it offered significant advantages over the conventional face-to-face style (Zamora-Antuñano et al., 2022).
However, most positive descriptions of online higher education might depend on technological promises rather than on real-life online higher educational practices based on didactics (Pittman, 2013). Focusing on the programmatic definitions of online higher education might fail to develop and practice the theory during the COVID-19 pandemic (Pittman, 2013). In addition, some studies have demonstrated that higher dropout rates have been known for some time during the COVID-19 pandemic. Students were frequently frustrated due to a lack of higher meta-cognitive skills, insufficient feedback from their instructors, and difficulties in distance learning (Tan et al., 2021). Moreover, some literature considers online higher education a passive agent, tarnishing higher education (García-Peñalvo et al., 2021). Thus, this study proposed a research question:
RQ4: Can online higher education be an active agent for changes?
2.5 Online technology and a sustainable education
Sustainable higher education refers to the ability of universities to meet the specific needs of industries. Universities might collaborate with the industry to fulfill its needs and expectations because almost all universities could not produce employable and job-ready graduates by themselves (Al-Tabbaa & Ankrah, 2018). Nevertheless, a university could not collaborate with industries because of economic, cultural, social, or time constraints. There was a need for a mechanism that was effective and better to fulfill the expectations of the industry (Al-Tabbaa & Ankrah, 2018). Meanwhile, a complete face-to-face delivery mode had its disadvantages. Online technology would deliver online higher education. Studies revealed that it would reduce or address the challenges through online technology (Al-Tabbaa & Ankrah, 2018). Some research revealed that online technology might effectively help deliver online teaching and learning, benefiting the society (Coccia, 2019). Online technology is also a helpful logistical resource for in-class study.
However, some studies were concerned with the negative impact of online technology for its insufficient substitute for classroom teaching (Stauss et al., 2018). Online technology should be delivered and used with other means in different situations (Stauss et al., 2018). Knowledge delivery and skills output needed practice and teaching (Stauss et al., 2018). The efficient delivery of knowledge and skills could be realized by means of different teaching tools. An online implementation might not be limited to online technology. It was risky to take a complete online education shift (García-Peñalvo et al., 2021). Sustainable education challenged students psychologically via online technology in a developing society. The problem might result in a major issue because students were unlikely to ask questions due to the cultural norm, forming a gap between them and teachers. The presence of online technology might widen the gap during the COVID-19 pandemic (García-Peñalvo et al., 2021). Furthermore, the lack of a strong participatory learning culture in emerging societies might threaten students’ academic success and job readiness (García-Peñalvo et al., 2021).
Undeniably, online technology-assisted sustainable education was generally associated with employment (Alam & Forhad, 2021). Sustainable education made graduates have employable skills and academic success, which prepared them for job readiness (Alam & Forhad, 2021). Students’ academic score was considered a predictor to identify the success of education. Although sustainable education might cater to market popularity, online technology could ensure that the right knowledge and skills were put first for different modes of technology occurring in different situations (Alam & Asimiran, 2021). Online technology could be useful for academic success, as it could easily make all students focus on online learning (Alam, 2021). However, as the COVID-19 pandemic continued to spread, there was some concern that only online technology-assisted sustainable education was favorable for students with special socio-economic status. Furthermore, it could not guarantee qualified and proper education. Thus, it could not be promised that graduates could be successful in academic career and job readiness via time consumption (Alam, 2021). Thus, this study proposed a research question:
RQ5: Does online technology provide a sustainable education comparison of academic success and job readiness before and during COVID-19?
Results
RQ4: Can online higher education be an active agent for changes?
4.4. An active agent
Given that the pandemic continues to spread and derail the normal operation, educational institutes adopted extensive online provisions through virtual learning platforms in some countries (Zamora-Antuñano et al., 2022). Online higher education is still the right choice for most students and is becoming increasingly popular as an attractive option for students during the COVID-19 pandemic. Although students’ dropout of online higher education could be high for many reasons, such as persistence and participation, online higher education still had increasingly fierce potentials (Zamora-Antuñano et al., 2022), which would enhance students’ educational experience. Furthermore, the COVID-19 pandemic helped online higher education gain market popularity and leverage, intentionally or unintentionally.
Overall, due to the paradigm change caused by the COVID-19 pandemic and the shift from traditional in-class education to online education, online higher education should be a promoter of development (Zamora-Antuñano et al., 2022). As an active agent, it also lay in its response to risks flexibly. It somehow succeeded in realizing more learning opportunities worldwide, which people had promised before. Moreover, in this new online higher education context, those with time, distance, and access difficulties were provided genuinely accessible learning opportunities. Moreover, as a real change agent, online higher education was a substantial return to the investment for universities (Zamora-Antuñano et al., 2022). Even the epidemic has accelerated online higher education to become an active agent.
RQ5: Does online technology provide a sustainable education comparison of academic success and job readiness before and during COVID-19?
4.5. The role of online technology
It was found that online technology-based sustainable education might be the main tool for making money (Alam & Asimiran, 2021). It could not ensure a sustainable education system for better education development (Managi et al., 2021). Online technology in the name of a sustainable education might mislead students for lack of a supply cycle and a specific demand (Alam & Asimiran, 2021). Online technology-based sustainable education was a tool to earn money. It would tarnish the reputation of higher education for a long time. Consequently, the role of sustainable education will be doubtful (Alam & Asimiran, 2021). Online technology would aggravate the diploma crisis during the COVID-19 pandemic (Alam & Asimiran, 2021). So a rigid and regulatory framework was needed to ensure the quality of sustainable education; the framework should be driven with application and required revision continuously during the COVID-19 pandemic. Some literature explored a better technique that might be employed to ensure a sustainable education between industry and universities via relieving the diploma disease crisis (García-Peñalvo et al., 2021).
Some studies pointed out that online technology received both pros and cons (Alam, 2021). Students in online education had higher grades than in the conventional face-to-face teaching method. Online technology might help improve students’ academic performance, although students’ academic achievements were doubtful, as there was a possibility to fabricate grading (García-Peñalvo et al., 2021). Online technology provided a lot of innovative ideas in many cases to improve sustainable education, yet pieces of evidence suggested that students performed well in job readiness before the COVID-19 pandemic (Alam, 2021). Furthermore, online technology had shortcomings in providing sustainable education during the COVID-19 pandemic (Alam, 2021). It might threaten sustainable education in some developing countries (Alam, 2021). The efficacy of online knowledge and skills delivery raised doubts as students with better grades did not correlate with job readiness and real competencies. Although the technology seemed to meet the requirements of sustainable education, it damaged its reputation and essential nature, such as creating grades (Alam, 2021). Graduates’ academic success might not reflect the realities of the job market. Students’ credentials or qualifications did not mean their enough job readiness.
5 Discussion
The study also finds that online higher education presents a series of advantages, such as the delivery of knowledge and skills, the application of techniques, and timely information management due to the transition period (Zamora-Antunano et al., 2021). Online higher education applied timely educational applications to meet the community needs (Zamora-Antunano et al., 2021). Online higher education uses the platform to make teachers and students exchange information. Furthermore, it produces a good environment for all participants (Rodriguez-Segura et al., 2020). The implementation of online higher education facilitated the teaching and learning process. In addition, students’ cognitive abilities develop better, and they are active participants in the process. Thus, their academic performance gets improved.
Online technology in providing sustainable education should not be a permanent option in students’ academic success and job readiness (Zamora-Antunano et al., 2021). People should not regard it as one of the mainstream education services for teachers and learners. Online technology is a flexible and convenient online learning way during the COVID-19 pandemic. It con-tributes to the delivery of knowledge and skills compared with the conventional learning model. However, strict and prudent measures are also required to ensure its benefits (Alam, 2021). Graduates’ academic performance may measure the effectiveness of a sustainable education (Stauss et al., 2018). However, there are no reciprocal connections between academic success and job readiness. Academic performance is only a main parameter to measure a sustainable educational effect (Jalongo, 2021). Somehow, the COVID-19 pandemic makes academic achievement overvalued. An online technology-based sustainable education of academic success and job readiness during the COVID-19 pandemic becomes dysfunctional. Overall, we should explore a comprehensive approach to a sustainable education via online technology during COVID-19.
6 Conclusions
Major findings
The study also attempted to explore some common perceptions regarding the role of online higher education and online technology. We have focused on the actual change and status of online higher education. We found that online higher education could be an active agent for changes, and online technology might partly provide a sustainable education comparison of academic success and job readiness before and during COVID-19. Generally, online higher education and technology benefited the society. Meanwhile, both of them could improve teachers’ and students’ perceptions. The COVID-19 pandemic severely damaged the economy worldwide, where online higher education could function to a certain degree.
Limitations
Although the study has tried to demonstrate some perceptions regarding the role of online higher education and online technology, we admit that it is difficult to know to what extent online higher education as an active agent is realized during the limited period. Moreover, its impact on higher education still needs time to demonstrate. In addition, we need to further explore whether online technology provides a sustainable education comparison of academic success and job readiness before and during COVID-19. This study adopts a qualitative method which can be enhanced by quantitative methods.
Future research directions
This study focused on teachers’ role, satisfaction, and digital literacy. Future research could explore the factors that hinder them through empirical experiments for the pandemic will most likely continue for some time. Future research could include more factors that might influence the educational outcomes during the pandemic. Future re-search could explore teachers’ intrinsic cognitive loads owing to paradoxical findings. Another research stream is to testify teacher satisfaction levels and investigate what policymakers can do in high-quality educator preparation. We might take the next plausible step to clearly answer these research questions via convincing methods. Future research might also carry out more case studies via representative samples and enough data.”
References:
- Alam, G. M. (2021). Does online technology provide sustainable HE or aggravate diploma disease? Evidence from Bangladesh-a comparison of conditions before and during COVID-19. Technology in Society, 66. Https://doi.org/10.1016/j.techsoc.2021.101677
- Alam, G. M., & Asimiran, S. (2021). Online technology: Sustainable higher education or diploma disease for emerging society during emergency-comparison between pre and during COVID-19. Technological Forecasting & Social Change, 172. Https://doi.org/10.1016/j.techfore.2021.121034
- Alam, G. M., & Forhad, M. A. R. (2021). Clustering secondary education and the focus on science: Impacts on higher education and the job market in Bangladesh. Comparative Education Review, 65(2), 310-331. Https:// doi.org/10.1086/713315
- Al-Tabbaa, O., & Ankrah, S., (2018). 'Engineered' University-Industry Collaboration: A Social Capital Perspective. European Management Review, 16(3), 543–565. Https://doi.org/10.1111/emre.12174
- Coccia, M., (2019). The theory of technological parasitism for the measurement of the evolution of technology and technological forecasting. Technological Forecasting & Social Change, 141(4), 289–304. Https://doi.org/10.1016/j.techfore.2018.12.012
- García-Peñalvo, F. J., Corell, A., Abella-García, V., & Grande-de-Prado M. (2021). Recommendations for mandatory online assessment in higher education during the COVID-19 pandemic. In D. Burgos, A. Tlili, & A Tabacco. (Eds.), Radical solutions for education in a crisis context. Lecture Notes in Educational Technology. Springer, Singapore. Https://doi.org/10.1007/978-981-15-7869-4_6
- Jalongo, M. R. (2021). The effects of COVID-19 on early childhood education and care: Research and resources for children, families, teachers, and teacher educators. Early Childhood Education Journal, 49, 763-774. Https://doi.org/10.1007/s10643-021-01208-y
- Lee, K. (2017). Rethinking the accessibility of online higher education: A historical review. Internet and Higher Education, 33, 15-23. Https://doi.org/10.1016/j.iheduc.2017.01.001
- Managi, S., Lindner, R., & Stevens, C. C., 2021. Technology policy for the sustainable development goals: From the global to the local level. Technological Forecasting and Social Change, 162(1). Https://doi.org/10.1016/j.techfore.2020.120410
- Pittman, V. (2013). University correspondence study: A revised historiographic perspective. In M. G. Moore (Ed.), Handbook of distance education (3rd ed., pp. 21‒37). New York, NY: Routledge.
- Rodriguez-Segura, L., Zamora-Antunano, M. A., Rodriguez-Resendiz, J., Altamirano-Corro, J. A., Paredes-Garcia, W. J., & Cruz-Perez, M. A. (2020). Teaching challenges in COVID-19 scenery: Teams platform-based student satisfaction Approach, Sustainability, 12(18). Https://doi.org/10.3390/su12187514
- Stauss, K., Koh, E., & Collie, M. (2018). Comparing the effectiveness of an online human diversity course to face-to-face instruction. Journal of Social Work Education, 54(3), 492-505. Https://doi.org/10.1080/10437797.2018.1434432
- Tan, K. H., Chan, P. P., & Said, N. E. M. (2021). Higher education students' online instruction perceptions: A quality virtual learning environment. Ssustainablity, 13(19). Https://doi.org/10.3390/su131910840
- Zamora-Antuñano, M. A., Rodríguez-Reséndiz, J., Cruz-Pérez, M. A., Reséndiz Reséndiz, H., Paredes-García, W. J., & Díaz, J. A. G. (2022). Teachers’ perception in selecting virtual learning platforms: A case of Mexican highereducation during the COVID-19 crisis. Sustainability, 14(195). Https://doi.org/10.3390/su14010195
- Zamora-Antunano, M. A., Rodriguez-Resendiz, J., Segura, L. R., Perez, M. A. C., Corro, J. A. A., Paredes-Garcia, W. J., & Rodriguez-Resendiz, H. (2021). Analysis of emergency remote education in COVID-19 crisis focused on the perception of the teachers. Sustainability, 13(7). Https://doi.org/10.3390/su13073820
In the analysis section, and especially in the limitations part, it would help a lot to give more arguments, and to include the answers to your research questions.
Response: Thank you for carefully and patiently reviewing of our manuscript. According to your advice, we conducted a series of supplementary to support our research. All additions in the revised manuscript have been marked in red.
Here are the details:
6.2. Limitations
The limitations are as follows.
There are some limitations to this study. First, it is not easy to precisely measure them in the context of such large-scale online education. Second, we tried to reveal the changes in teachers’ role, satisfaction levels, and their digital literacy development. Due to the length of teaching content and students’ online learning behavior in the online teaching-learning process, it is complicated to accurately investigate the changes in teacher professional role, career satisfaction levels, and the real need for their digital literacy during the COVID-19 pandemic. Third, it is necessary to provide sufficient evidence to support the study. It is hard to gain sufficient support due to the inadequate resources. In addition, we cannot exhaust all samplings even if we have a large size of samplings and enough databases since some articles may elude our search. So the reliability of results might be lowered. Moreover, the results of various studies are inconsistent. Some empirical studies concluded that the COVID-19 pandemic enhanced teacher professional role and satisfaction level, and teachers ‘digital literacy was well developed, while others revealed the reverse results.
Although the study has tried to demonstrate some perceptions regarding the role of online higher education and online technology, we admit that it is difficult to know to what extent online higher education as an active agent is realized during the limited period. Moreover, its impact on higher education still needs time to demonstrate. In addition, we need to further explore whether online technology provides a sustainable education comparison of academic success and job readiness before and during COVID-19. This study adopts a qualitative method which can be enhanced by quantitative methods.
On line 311-317 it mentions that the professional roles changed in a complicated way, so that more activities for the teaching process were assigned. Implying this as decrease in teacher satisfaction, requires broader argument enrich and clarify some unclear aspects of the analysis.
It is better to provide a more up-to-date bibliographic review. There are excellent current examples published of the subject in the Information Journal.
The article has no reference from the journal Information. Journal Information has excellent references related to the topic.
I think it will be a good contribution after making these adjustments.
Response: Thank you for carefully and patiently reviewing of our manuscript. According to your advice, we have added some literature to support our research. All additions in the revised manuscript have been marked in red.
Here are the details:
“The research on the improvement of teachers' skills and affordance of the necessary conditions in the digital era has become a hot issue (Jesmin & Ley, 2020).”
“Overall, these studies highlight the importance of teacher career satisfaction for future research and methodological diversity (Batista et al., 2021).”
References:
Jesmin, T., & Ley, T. (2020). Giving teachers a voice: A study of actual game use in the classroom. Information, 11(1), 55. Https://doi.org/10.3390/info11010055
Batista, J., Santos, H., & Marques, R. P. (2021). The use of ICT for communication between teachers and students in the context of higher education institutions. Information, 12(11), 479. Https://doi.org/10.3390/info12110479

Reviewer 3 Report
It is a well-developed and pertinent study in the face of the main global emergency. It is a relevant work at this moment. However, it needs some improvements before publishing.
The subject of the study on satisfaction, the role and digital literacy of the teacher are well developed, but it would be good to clarify these concepts from the introduction for better clarity.
I like to read the conceptual framework; however, I like to see the link between the role of education and the interference of virtual learning in this role before outlining the goal, the objectives, and the research questions. Also, they must be aligned.
The author can get use the sources at ‘sustainable education for society the emergency: comparison between technological forecasting and social change before and during COVID-19’. I consider the discussion as the main opportunity of the article if it includes the objectives in the introduction, and integrates the questions:
Can online higher education be an active agent for change?
Does online technology provide a sustainable education comparison of academic success and job readiness before and during COVID-19?
Therefore, the author or authors should review the most recent articles to improve their research
- https: // doi. org / 10.3390 / su14010195
- https://doi.org/10.3390/su13073820
- https://doi.org/10.3390/su12187514
It would be good to strengthen some arguments in the results section.
In the analysis section, and especially in the limitations part, it would help a lot to give more arguments, and to include the answers to your research questions.
On line 311-317 it mentions that the professional roles changed in a complicated way, so that more activities for the teaching process were assigned. Implying this as decrease in teacher satisfaction, requires broader argument enrich and clarify some unclear aspects of the analysis.
It is better to provide a more up-to-date bibliographic review. There are excellent current examples published of the subject in the Information Journal.
The article has no reference from the journal Information. Journal Information has excellent references related to the topic.
I think it will be a good contribution after making these adjustments.
Author Response
Response to reviewer comments
Dear reviewer,
Thank you very much for your careful review work. We have conducted substantial revisions based on your constructive suggestions as follows.
Please feel free to contact us if you have any other questions.
Best regards!
Sincerely yours,
Dr. Li Ming
Prof., Dr. Yu Zhonggen
Beijing Language and Culture University
Response to Reviewer 3:
- Typo: “It is beneficial to improve learning and teachING effectively”
Response: Thank you for carefully and patiently reviewing of our manuscript. According to your advice, we have revised this sentence in the revised manuscript.
Here are the details:
“It is beneficial to improve learning and teaching effectively,”
- I believe that there is a need to better conceptualize the practice of teaching and Covid19. The following references might e helpful if the authors find them relevant:
- Online university teaching during and after the Covid-19 crisis: Refocusing teacher presence and learning activity. Postdigital Science and Education, 2,923–945. https://doi.org/10.1007/s42438-020-00155-y
- Teaching in the age of Covid-19 - A longitudinal study. Postdigital Science and Education, 3(3), 743–770. https://doi.org/10.1007/s42438-021-00252-6
- Adapting to online teaching during COVID-19 school closure: teacher education and teacher competence effects among early career teachers in Germany. European Journal of Teacher Education, 43(4), 608-622. https://doi.org/10.1080/02619768.2020.1809650
Response: Thank you for your constructive suggestion. According to your advice, we have added the literature to our paper in the revised manuscript.
Here are the details:
“The better teachers' digital literacy development, the more competent teachers are in education (Rapanta et al., 2020).”
“It seemed that digital competence had become a necessity, not a need, in education (Jandrić et al., 2021).”
“Current studies showed that digital literacy played a leading role in high-quality education in current situations (Konig et al., 2020).”
References:
Rapanta, C., Botturi, L., Goodyear, P. & Guàrdia, L., Koole, M., (2020). Online university teaching during and after the Covid-19 crisis: Refocusing teacher presence and learning activity. Postdigital Science and Education, 2, 923–945 Https://doi.org/10.1007/s42438-020-00155-y
Jandrić, P., Bozkurt, A., McKee, M., & Hayes, S. (2021). Teaching in the age of Covid-19 - A longitudinal study. Postdigital Science and Education, 3(3), 743–770. Https://doi.org/10.1007/s42438-021-00252-6
Konig, J., Jager-Biela, D. J., & Glutsch, N. (2020). Adapting to online teaching during COVID-19 school closure: Teacher education and teacher competence effects among early career teachers in Germany. European Journal of Teacher Education, 43(4), 608-622. Https://doi.org/10.1080/02619768.2020.1809650
- For Figure 1, authors may adopt PRISMA framework.
Response:Thank you for your comments. We have adopt PRISMA framework, and we detailed the design process and methodology of the study and added more requirement component in the revised manuscript.
Here are the details:
- “Data analysis - Twenty-one publications were coded as depicted to answer the research questions in 179 Table 1, as the primary citations were included to be suitable for inclusion.” This section can be expanded and the authors may provide more explanation. Besides, two lines and one single sentence does not constitute a full paragraph.
Response: Thank you very much for your constructive suggestions. We have carefully revised and made some supplementary to support our research as follows:
3.3 Data appraisal and analysis
We use critical appraisal tools from The University of West England Framework [1, 2]. We refined the results and scored each selected paper to get high-quality studies with STARLITE by removing the irrelevant and lower-quality results [2, 3].
Two reviewers included and excluded the literature. If they could not reach an agreement on any decision, a third reviewer would be invited to finalize it. The inter-rater reliability reaches a higher level (k = 0.93). Inter-rater reliability has used the formula suggested by Miles and Huberman [73]. We screened articles in the following steps using the inclusion criteria listed in Figure 1. The articles were selected with the inclusion criteria shown in Table 1. Specifically, we implemented the inclusion and exclusion based on nine steps.
Step 1: We searched the Web of Science database in November 2021 and restricted the publishing year to 2020-2021, resulting in 662 articles.
Step 2: We excluded non-English articles, leading to 637 articles.
Step 3: We read the titles and abstracts (some literature had no abstracts) and ex-cluded 483 articles that were irrelevant to the search string or didn’t belong to educational areas (education, computer, language, linguistics, literature, mathematic, music, chemistry, sociology, history, geography, biology, and physics, etc.).
Step 4: We filtered an editorial collection, reports, or review articles, leading to 45 articles.
Step 5: We excluded the duplicates and measured the sample size and sampling strategy. It resulted in 18 articles.
Step 6: We filtered articles short of online teaching platforms, leading to 16 articles.
Step 7: We used a non-systematic hand search after the first searching round to avoid missing additional articles. The hand search resulted in 2 articles that met the inclusion criteria.
Step 8: In December 2021, we conducted the third round search using the original keyword across the Web of Science with the same criteria, and there was little difference. We screened the titles and abstracts, resulting in the addition of 3 articles.
Step 9: We used a STARLITE appraisal and analyzed the twenty-one articles.
References:
- Booth, A. (2006). “Brimful of STARLITE” toward standards for reporting literature searches. Journal of the Medical Library Association, 94, 421–429.
- Miles, M. B., & Huberman, A. M. (1994). Qualitative data analysis: An expanded
Sourcebook (2nd ed.). Thousand Oaks: Sage Publications.
- Moule, P., Pontin, D., Gilchrist, M., & Ingram, R. Critical appraisal framework. Available online: Http://learntech.uwe.ac.uk/da/Default.aspx?pageid=1445 (accessed on 22 December 2021)
- Yu, Z. G., Gao, M. L., & Wang, L. F. (2020). The effect of educational games on learning outcomes, student motivation, engagement and satisfaction. Journal of Educational Computing Research, 59, 522-546. Https://doi.org/10.1177/0735633120969214
- The discussion section is too weak. Please compare and contrast your finding by benefiting from the related literature. As is, it does not stand like a discussion section.
Response: Thank you very much for your suggestion. To illustrate this part more clearly, we have added the following to the manuscript.
Here are the details:
- Discussion
“Almost all teaching staff coped with the sudden shift from the face-to-face teaching model to online teaching during the COVID-19 pandemic. They must struggle to adapt to the additional stresses and workloads (Rapanta et al., 2020). Accordingly, it caused the changes in teachers' professional role, declination of their satisfaction level, and many digital literacy challenges because teachers must balance responsibilities, teaching, work-life, and development during the urgent imperative to online teaching. Teachers had more obligations, which often entailed teachers facing greater psychological pressure. In addition, pressure can also result from a lack of technical support. There are still teachers with no or little digital literacy in online teaching as they come from all ages and backgrounds.”
Reference:
Rapanta, C., Botturi, L., Goodyear, P. & Guàrdia, L., Koole, M., (2020). Online university teaching during and after the Covid-19 crisis: Refocusing teacher presence and learning activity. Postdigital Science and Education, 2, 923–945 Https://doi.org/10.1007/s42438-020-00155-y
- Again, “Future research directions” section is too weak. What would you suggest to different stakeholders and the implications for future research would be?
Response: Thank you very much for your positive comments and constructive suggestions. According to your advice, we conducted some supplementary to this section in the revised manuscript.
Here are the details:
“6.3. Future research directions
This study focused on teachers’ role, satisfaction, and digital literacy. Future research could explore the factors that hinder them through empirical experiments for the pandemic will most likely continue for some time. Future research could include more factors that might influence the educational outcomes during the pandemic. Future re-search could explore teachers’ intrinsic cognitive loads owing to paradoxical findings. Another research stream is to testify teacher satisfaction levels and investigate what policymakers can do in high-quality educator preparation. We might take the next plausible step to clearly answer these research questions via convincing methods. Future research might also carry out more case studies via representative samples and enough data.”

Reviewer 4 Report
- Typo: “It is beneficial to improve learning and teachING effectively”
- I believe that there is a need to better conceptualize the practice of teaching and Covid19. The following references might e helpful if the authors find them relevant:
- Online university teaching during and after the Covid-19 crisis: Refocusing teacher presence and learning activity. Postdigital Science and Education, 2,923–945. https://doi.org/10.1007/s42438-020-00155-y
- Teaching in the age of Covid-19 - A longitudinal study. Postdigital Science and Education, 3(3), 743–770. https://doi.org/10.1007/s42438-021-00252-6
- Adapting to online teaching during COVID-19 school closure: teacher education and teacher competence effects among early career teachers in Germany. European Journal of Teacher Education, 43(4), 608-622. https://doi.org/10.1080/02619768.2020.1809650
- For Figure 1, authors may adopt PRISMA framework.
- “Data analysis - Twenty-one publications were coded as depicted to answer the research questions in 179 Table 1, as the primary citations were included to be suitable for inclusion.” This section can be expanded and the authors may provide more explanation. Besides, two lines and one single sentence does not constitute a full paragraph.
- The discussion section is too weak. Please compare and contrast your finding by benefiting from the related literature. As is, it does not stand like a discussion section.
- Again, “Future research directions” section is too weak. What would you suggest to different stakeholders and the implications for future research would be?
Author Response

(The authors gave the same response as above.)

Round 2
Reviewer 2 Report
Changes and suggestions have been addressed by the authors. I believe the article has improved with the changes made.
My congratulations to the authors
Reviewer 4 Report
Thank you for revising the manuscript based on the suggestions provided. It looks in a good shape and I believe that it will contribute to the related literature.